# Boron Nitride/Polyurethane Composites with Good Thermal Conductivity and Flexibility

**DOI:** 10.3390/ijms24098221

**Published:** 2023-05-04

**Authors:** Xinze Yang, Jiajing Zhang, Liangjun Xia, Jiahao Xu, Xuenan Sun, Chunhua Zhang, Xin Liu

**Affiliations:** 1College of Material Science and Engineering, Wuhan Textile University, Wuhan 430200, China; 15623622307@163.com (X.Y.); 2215043052@mail.wtu.edu.cn (J.X.); 2State Key Laboratory of New Textile Materials and Advanced Processing Technologies, Wuhan Textile University, Wuhan 430200, China; zjj18672519077@163.com (J.Z.); liangjun_xia@wtu.edu.cn (L.X.); 13037125505@163.com (X.S.); 3College of Textile Science and Engineering, Zhejiang Science and Technology University, Hangzhou 310016, China

**Keywords:** boron nitride, polyurethane, thermal conductivity, mechanical properties

## Abstract

Thermal insulating composites are indispensable in electronic applications; however, their poor thermal conductivity and flexibility have become bottlenecks for improving device operations. Hexagonal boron nitride (BN) has excellent thermal conductivity and insulating properties and is an ideal filler for preparing thermally insulating polymer composites. In this study, we report a method to fabricate BN/polyurethane (PU) composites using an improved nonsolvent-induced phase separation method with binary solvents to improve the thermal performance and flexibility of PU. The stress and strain of BN60/PU are 7.52 ± 0.87 MPa and 707.34 ± 38.34%, respectively. As prepared, BN60/PU composites with unordered BN exhibited high thermal conductivity and a volume resistivity of 0.653 W/(m·K) and 23.9 × 10^12^ Ω·cm, which are 218.71 and 39.77% higher than that of pure PU, respectively. Moreover, these composite films demonstrated a thermal diffusion ability and maintained good integrity after 1000 bending cycles, demonstrating good mechanical and thermal reliability for practical use. Our findings provide a practical route for the production of flexible materials for efficient thermal management.

## 1. Introduction

Flexible thermal conductive composites have the advantages of a light weight, simple processing, and a low manufacturing cost [1], which are crucial to addressing the issue of equipment overheating and have become the focus of scientific research [2]. However, most polymers have low thermal conductivity (TC; typically less than 0.3 W/(m·K)) [3]. A promising way to produce thermally conductive materials for industrialization is to make the composite material form a heat-conducting network and improve TC by adding an appropriate amount of thermally conductive filler. Hexagonal boron nitride (BN) is an inorganic ceramic material with a microstructure similar to that of graphene. It is an ideal thermally conductive filler for polymer composites because of its high TC of 390 W/(m·K) [4,5].

The thermal conductivity of polymer composites mainly depends on the thermal conduction chain and network formed by the filler. The formation of these networks is closely related to the structure, morphology, filling content of particles, and the interaction between filler and polymer matrix [6]. The commonly used structural design and processing of filled thermally conductive composite materials primarily include 3D printing technology, freeze-drying, cold-press sintering, hot-pressing, magnetic induction, ice templating, self-assembly, hybrid filler fabrication, nonsolvent induced phase separation, solvent evaporation, and melting methods [7,8,9,10,11,12,13,14].

Bozkurt et al. [15] studied PEI composites filled with different content of h-BN by a twin-screw extruder. They stated that the rheological behavior could be correlated to the thermal conductivity of composites. They determined the best dispersion state of h-BN according to storage modulus. Finally, with the optimized process conditions, the enhanced thermal conductivity of PEI were achieved. Jang et al. [16] regulated the distribution and alignment of h-BN in epoxy matrix through self-assembled core-shell e-BN microspheres. By shortening the heat-transfer pathways and increasing the contact surface area between the fillers, the composites with high thermal conductivity were obtained. Su et al. [17] designed a novel spherical hybrid filler and constructed the continuous three-dimensional thermal conduction network through melting mixing and hot compression. This continuous thermal conduction network reduced the interface thermal resistance of the matrix. When the content of filler was 30%, the thermal conductivity of the composite material increased by 6 times compared with that of pure polymer. Yu et al. [13] fabricate flexible and insulating elastomer-based 2D composite film with excellent TC through ball milling and hot-pressing. The hot-compression induced the alignment of h-BN and strong interactions in the polymer matrix, improving the thermal conductivity of the composites. Therefore, the process parameters and methods are critical to the construction of the overall thermal conductivity network of polymer composites.

Polyurethane (PU) has excellent mechanical properties, good insulation properties, and low-temperature resistance; however, its low TC limits its applications in electronic devices. Many investigations showed that although a large amount of BN can improve the thermal conductivity of PU, it will significantly deteriorate its strain and affect the practical application [18,19,20]. Based on our previous studies, the combination of the nonsolvent-induced phase separation (NIPS) method with binary solvents is beneficial for improving the interface properties of the filler and matrix [21,22,23]. Herein, binary solvents were employed to improve the interfacial interaction between the unmodified BN and PU matrix, and BN/PU composite films were fabricated using the improved NIPS method. Compared with BN, a boron nitride nanosheet (BNNS) has higher thermal conductivity, but it needs to be peeled off through various methods, such as ball milling or ultrasonic-assisted exfoliation with different surface modifier. Although these approaches yield relatively high quality BNNSs, the low production and high cost significantly restrict their scalability [24,25]. Based on the principle of simplicity and mass production, the intercalated BN is selected for this study. The influence of BN content on the structure, morphology, mechanical properties, thermal conductivity, and insulation performance of PU was analyzed.

## 2. Results and Discussion

### 2.1. Characterization of BN

XRD and FTIR analyses were performed to determine the composition of the particles. As shown in Figure 1b,c, the XRD pattern of the BN reveals the characteristic diffraction peaks at 26.7, 41.7, 43.9, 50.2, and 55.1°, which correspond to the (002), (100), (101), (102), and (004) planes, respectively [26]. The FTIR spectra of BN show peaks at approximately 780 and 1380 cm^−1^, respectively, corresponding to the deformation and tensile vibration of the B-N bond [27,28]. Figure 1d shows that the BN have a disk-like shape and diameters of 1~2 μm. Furthermore, the stability of the BN60 and BN60/PU solutions were analyzed as shown in Figure 1e. Compared with BN60, BN60/PU solution exhibits higher stability after 24 h, which can be attributed to the steric stabilization of BN by the PU chains. Photographs of the as-prepared PU and BN60/PU composite film are presented in Figure 1f. The BN60/PU composite film turns from transparent to white upon the incorporation of unmodified BN, which is attributed to the white color of scattered BN particles (photograph inserted in Figure 1d).

### 2.2. Structure of BN/PU Composite Films

XRD and FT-IR were performed as effective tools to determine the chemical composition and structure of the PU and BN/PU composite films.

In Figure 2a, the broad diffraction peak located at 2θ = 20° in the PU and BN/PU composites is attributed to the amorphous structure of PU. The characteristic crystallization peak of BN appears in the XRD pattern of BN/PU composite, indicating that the NIPS process of the BN and PU did not change the crystal structure of BN and good TC could be maintained. The intensity of these diffraction peaks increase with increasing BN content, indicating that PU chains and the processing approach do not intercalate BN. The orientation of BN in PU has a significant influence on thermal conductivity. XRD was used to further investigate the degree of orientation of BN in the PU.

The orientation of BN sheets in the polymer matrix is one of the essential factors affecting the thermal conductivity, and it can be characterized by an XRD analysis. The degree of the through-plane orientation (TPO) of BN/PU composite films was calculated, and the equation can be described as follows [29]:TPO = I_100_/(I_100_ + I_002_) × 100%
where I_100_ and I_002_ are the intensities of the (100) and (002) planes of the BN, respectively. The (100) plane relates to the vertically aligned BN sheets and the (002) plane represents the horizontal ones [29].

The TPO values of BN40/PU, BN50/PU, BN60/PU, and BN70/PU composite films are 8.64, 8.03, 7.33, and 7.77%, respectively. With the increase of BN content, the TPO of composite film showed a slight decreasing trend, indicating that the horizontal orientation of BN increase. However, compared with other reported composites, the TPO value of the BN/PU composites is significantly lower, indicating that the unmodified BN are still randomly oriented in PU [27,28,29]. The disorderly arrangement of BN is beneficial for improving the isotropic heat transfer in composites, which can be used in electronic packaging materials.

The FT-IR spectra of PU and BN/PU composite films are presented in Figure 2b. The characteristic absorptions of the PU were observed at 3312 cm^−1^ (N-H stretching vibration), 2939 cm^−1^ (asymmetric stretching vibration of -CH_2_-), 2853 cm^−1^ (symmetric stretching vibration of -CH_2_-), 1730 cm^−1^ and 1700 cm^−1^ (stretching vibration of C=O), 1531 cm^−1^ (N-H bending vibration), 1413 cm^−1^ (bending vibration of -CH_2_ and -CH_3_), 1222 cm^−1^ (C-N stretching vibration), and 1075 cm^−1^ (stretching vibration of C-O-C) [30]. By adding BN, the peak of BN40/PU, BN50/PU, BN60/PU, and BN70/PU composite films at 3312 cm^−1^ shifted to a high wavenumber, indicating the formation of a hydrogen bond between PU and BN, which may be due to the introduction of a hydroxyl group in the solution preparation process. In the spectra of the BN/PU composite films, new peaks at 1372 and 765 cm^−1^ appeared in the BN/PU composite films, which belonged to the characteristic peaks of BN. Additionally, no other new peaks appear, indicating that the interactions between BN and PU were physical.

### 2.3. Mechanical Properties of BN/PU Composite Films

As the mechanical properties of the composites also play a crucial role in thermal management materials, the stress and strain of the BN/PU composite film were obtained, as shown in Figure 3. With an increase in filler content, the stress and strain of the BN/PU composite film decreased. This is because the intrinsic BN is brittle relative to PU, and the increased BN content breaks the continuous network of PU. In addition, the accumulated BN and local stress concentrations can deteriorate the mechanical properties of the composite. As depicted in Figure 3, the stress and strain of BN60/PU are 7.52 ± 0.87 MPa and 707.34 ± 38.34%, respectively. When the BN content continued to increase to 70 wt%, the stress and strain of the BN70/PU are 5.00 ± 0.50 MPa and 274.67 ± 44.74%, respectively, and the strain of the composite film decreased significantly.

### 2.4. Thermal Properties of BN/PU Composite Films

The TG analyses of PU and BN/PU composite films were carried out to determine the thermal stability, and the related data were shown in Table 1. The temperature at 5% and 20% mass loss was defined as T_5%_ (initial decomposition temperature) and T_20%_ (main decomposition temperature). According to the TGA results of PU, T_5%_ and T_20%_ were 281.5 and 313.5 °C, respectively, and about 2.30% of the residue was retained. The BN/PU composite films with 40, 50, and 60 wt% BN loading exhibited a residual weight of 35.38, 49.12, and 60.002%, indicating that the BN is evenly distributed in the PU matrix (40~60% particle content). However, the residual mass of BN70/PU is only 64.92%, implying that the high content of BN is severely agglomerated, which disturbs the network of PU and makes the stress concentration. During stress loading, the BN aggregates break down and the strain of the composite film decreases significantly. Additionally, the initial decomposition temperature and the main decomposition temperature of BN/PU composite films were increased to varying degrees with the introduction of BN, indicating that the stability of the composite films was enhanced. Two factors can explain this: (i) the BN restricts the movement of PU chains and delays the thermal decomposition of PU; (ii) the formed thermal conduction pathway facilitates the transfer of heat and reduces local heat accumulation.

### 2.5. TC of BN/PU Composite Films

Figure 4a show the TC and TC enhancement of different BN/PU composite films. The pure PU exhibits a low TC of 0.170 W/(m·K), owing to the severe phonon scattering resulting from its low crystallinity. As shown in Figure 4b, the TC of the BN/PU composites increases with an increase in the BN content. The increase in the loading of particles increased the contact between BN, enhancing the probability of forming efficient thermally conductive pathways. The TC of BN60/PU and BN70/PU is 0.545 and 0.653 W/(m·K), which increases by 218.71 and 281.87% compared with PU.

With the development of wearable devices, higher requirements have been proposed for the flexibility of thermally conductive materials. Therefore, composites with excellent TC and mechanical properties are necessary for practical applications. Considering the comprehensive properties of the composites, BN60/PU, with excellent TC and mechanical properties, was selected for further study.

Furthermore, the mechanical properties and TC of the as-prepared BN/PU composite films with unmodified BN prepared by improved NIPS and traditional process methods (SE and NIPS) was compared. As shown in Figure 4d, the stress of BN60/PU were 307.59 and 175.46% higher than those of BN60/PU-NIPS and BN60/PU-SE, respectively. In addition, the BN60/PU strain was 1466.12 and 9122.16% higher than the strain of the composites prepared using the traditional NIPS and SE methods with the 60% unmodified BN. As depicted in Figure 4c, the TC of BN60/PU, BN60/PU-NIPS, and BN60/PU-SE are 0.545, 0.319, and 0.425 W/(m·K), respectively. The TC of BN60/PU are 70.85 and 28.24% higher than those of BN60/PU-NIPS and BN60/PU-SE composite film. Thus, the improved film formation method can improve the interfacial interaction between the BN and PU matrix, which is conductive to the formation of an effective thermal conduction network.

### 2.6. Morphology of BN60/PU Composite Film

The morphologies of the BN60/PU composite film were observed using SEM. As shown in Figure 4a, the freeze-fractured cross-sections of the BN60/PU composite film exhibit rough morphologies owing to the presence of BN. The green marks represent the distribution and agglomeration of BN in PU. In Figure 5, the BN was dispersed randomly in the PU matrix, which was consistent with the XRD analysis. Most of BN is closely embedded in the PU matrix, and small voids between BN and PU (indicated by the green arrow) are shown in Figure 5a. In addition, numerous sites for connecting BN were formed, effectively weakening the interference from phonon scattering and the interface thermal resistance. After the tensile tests, fewer voids were formed inside the BN60/PU, indicating a suitable interface (Figure 5b). This phenomenon suggests that a tougher interface was formed through NIPS using the binary solvents. Elemental mapping of B further confirmed the connected heat conduction network (Figure 5c).

### 2.7. Thermal Management Capability and Insulation of BN60/PU Composite Film

To visually demonstrate the thermal management applications of the BN60/PU composite film, an infrared thermal camera was used to record the heat transfer process. First, the samples were placed in a hot atmosphere at 80 °C for 30 min to ensure a uniform temperature. Then, it was transferred to a foam stage at laboratory temperature and captured at 0, 10, 20, 30, 60, 90, 120, 150, and 180 s.

As shown in Figure 6a, the surface temperature of each sample decreased with time. The surface temperature of BN60/PU exhibited the fastest rate of decrease, indicating that it had a good thermal diffusion ability. For example, after 30 s, the surface temperatures of PU and BN60/PU were 37.8 and 28.0 °C, respectively.

Furthermore, a thermal and cool cycling test was conducted to evaluate the potential damage attributed to temperature. The temperature of BN60/PU during the heating and cooling cycles is shown in Figure 6b,c. During the entire process of 40 cycles, the temperature of the composite was stable, indicating the good thermal stability of the BN60/PU composite. Additionally, Figure 6b indicates that the composite film still has good thermal conductivity after 48 h of standing at low temperature.

The mechanical stability of the composite film is crucial during thermal cycling. The occurrence of cracks deteriorates the thermal and mechanical performances of the composites. Figure 6c shows the flexibility of composite film. Moreover, the BN60/PU maintained good integrity and showed a negligible decrease in the surface temperature after 1000 cycles of bending, thus demonstrating good mechanical and thermal reliability in practical use (Figure 6d).

In the practical application of thermal management materials, insulativity is another important parameter. As shown in Figure 6e, the electrical insulation of BN60/PU is up to 23.9 × 10^12^ Ω·cm, which is 39.77% higher than that of PU, far exceeding the standard of electrical insulation (10^9^ Ω·cm), demonstrating the potential of BN60/PU in a variety of electronic applications.

## 3. Materials and Methods

### 3.1. Materials

Hexagonal boron nitride (BN, 1–2 μm) was purchased from Macklin Biochemical Co., Ltd. (Shanghai, China). The thermoplastic polyurethane (PU, 1185A) was provided by BASF Co., Ltd. (Ludwigshafen, Germany). N,N-dimethylformamide (DMF, 99.5 %, AR) and toluene (TL, 99.5 %, AR) were purchased from Sinopharm Chemical Reagent Co., Ltd. (Shanghai, China). All chemicals were used directly without further treatment. The coagulating bath was made with deionized water (DI water, conductivity ≤ 16 MΩ·cm).

### 3.2. Fabrication of Composites

Figure 1a illustrates the preparation of BN/PU composite films using the improved nonsolvent-induced phase separation (NIPS) process using binary solvents. First, the calculated BN and PU were dissolved in binary solvents (DMF:TL = 1:1 wt%) to prepare a 30 wt% casting solution. Second, the mixture solution was cast onto a clean glass plate to prepare film. The film was quickly moved into water coagulation bath for 120 min. Then, the formed film was dried at 80 °C for 1 h. Finally, the as-prepared film was pressed under 5 MPa pressure at 100 °C for 3 min. The PU with 0, 40, 50, 60, and 70 wt% BN content were coded as PU, BN40/PU, BN50/PU, BN60/PU, and BN70/PU, respectively.

The PU filled with 60 wt% BN formed by traditional NIPS method was named BN60/PU-NIPS according to previous publication [31]. The PU was first dissolved in DMF, and the mixture was cast on a glass plate and immersed in DI water for 120 min. After that, the formed film was dried at 80 °C for 1 h. Finally, the composite film was pressed under 5 MPa pressure at 100 °C for 3 min.

The PU filled with 60 wt% BN formed by traditional solvent evaporation (SE) method was named BN60/PU-SE [32]. The composite film was prepared by the following steps. PU was dissolved in DMF, and the mixture was cast on a glass plate and dried at 80 °C for 1 h. After that, the formed film was pressed under 5 MPa pressure at 100 °C for 3 min.

### 3.3. Characterization

The morphology and microstructures of all samples studied were measured with a scanning electron microscope (SEM, Phenom pure, Thermo Scientific, Waltham, MA, USA) and a field emission scanning electron microscope (FESEM, Zeiss sigma300, Jena, Germany) with energy dispersive spectrometer (EDS, Oxford X-MAX, Oxford, UK). The crystalline structure of BN particles and different BN/PU composites were identified with X-ray diffraction (XRD, Empryrean, Almelo, The Netherlands) with a Cu Kα radiation at 40 kV and 40 mA. The chemical compositions of BN particles and different BN/PU composites were recorded with Fourier transform infrared (FT-IR, NICOLET IS-50, Thermo Scientific) in the range of 4000–600 cm-1. The stress–strain curves of composites were measured using Instron 5967 at a speed of 20 mm/min. Thermogravimetric (TG) analysis was performed on an TG209F instrument at a heating rate of 10 °C/min from laboratory temperature to 800 °C under nitrogen atmosphere. The thermal conductivity (TC) of each sample was measured with thermal conductivity instrument (DTC-300, TA Instruments, New Castle, DE, USA) according to ASTM D5470. The temperature of the composite was recorded with an infrared thermograph (HIKMICRO, Shaanxi, China). The electrical insulation was measured with a high-resistance meter (TH2683, Changzhou Tonghui Electronic Co. Ltd., Changzhou, China).

## 4. Conclusions

A BN/PU composite with good TC and flexibility was fabricated using an improved NIPS method. An FTIR analysis showed that the addition of BN did not change the main structure of PU. An XRD analysis confirmed that the BN60/PU exhibited good horizontal orientation in PU; however, it was still disordered. TG results indicated that the BN improved the thermal stability of PU. Compared with PU, the initial decomposition temperature and main decomposition temperature of the BN60/PU composite film were increased by 21 and 46.9 °C, respectively. The TC of the prepared BN60/PU composite reached 0.653 W/(m·K), which is an increase of 218.71% compared with that of pure PU. Simultaneously, the strain of the BN60/PU was 1466.12 and 9122.16 % higher than the strain of the composites prepared with traditional SE and NIPS methods with the 60 wt% unmodified BN. Moreover, BN60/PU exhibits flexibility and good mechanical and thermal reliability. Additionally, the electrical insulation of BN60/PU is up to 23.9 × 10^12^ Ω·cm, far exceeding the standard of electrical insulation (10^9^ Ω·cm). This study provides a route to developing composites with good thermal conductivity, flexibility, and stability for potential applications in the thermal management field.

## Figures and Tables

**Figure 1 ijms-24-08221-f001:**
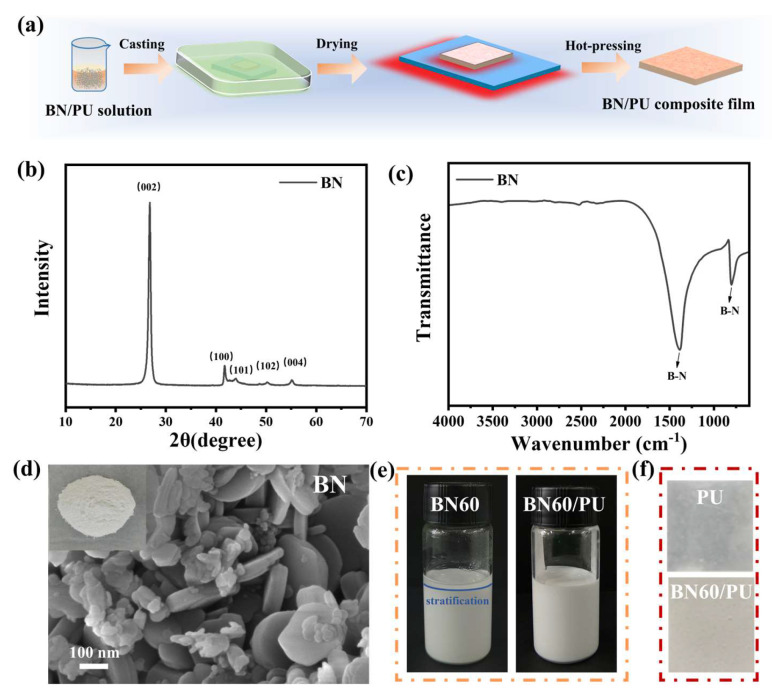
(**a**) Schematic illustration for preparation of BN/PU composite film; (**b**) XRD pattern and (**c**) FTIR spectra of BN particles; (**d**) SEM image of BN powder. Insert: photograph of BN powder; (**e**) Photographs of the BN60 and BN60/PU solutions after standing for 24 h; (**f**) photographs of the PU and BN60/PU composite film.

**Figure 2 ijms-24-08221-f002:**
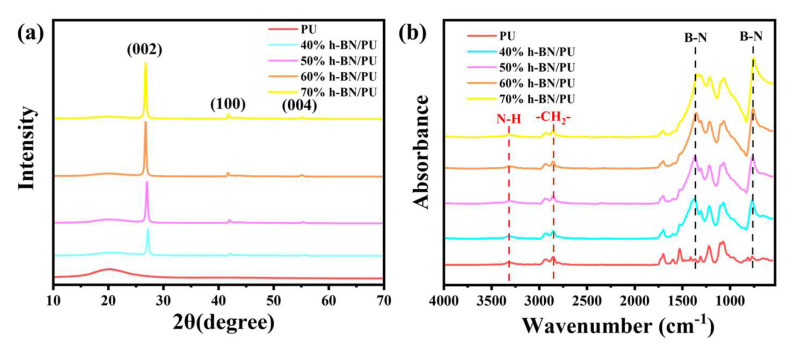
(**a**) XRD patterns and (**b**) FT-IR spectra of BN/PU composite films.

**Figure 3 ijms-24-08221-f003:**
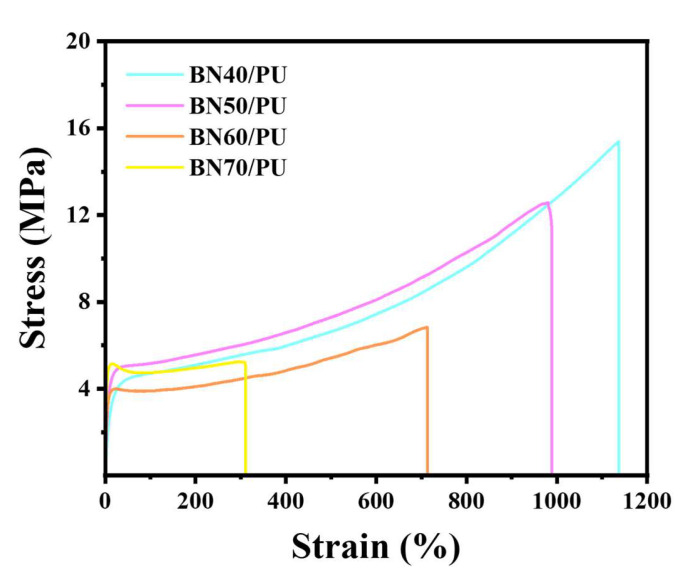
Stress–strain of BN/PU composite film.

**Figure 4 ijms-24-08221-f004:**
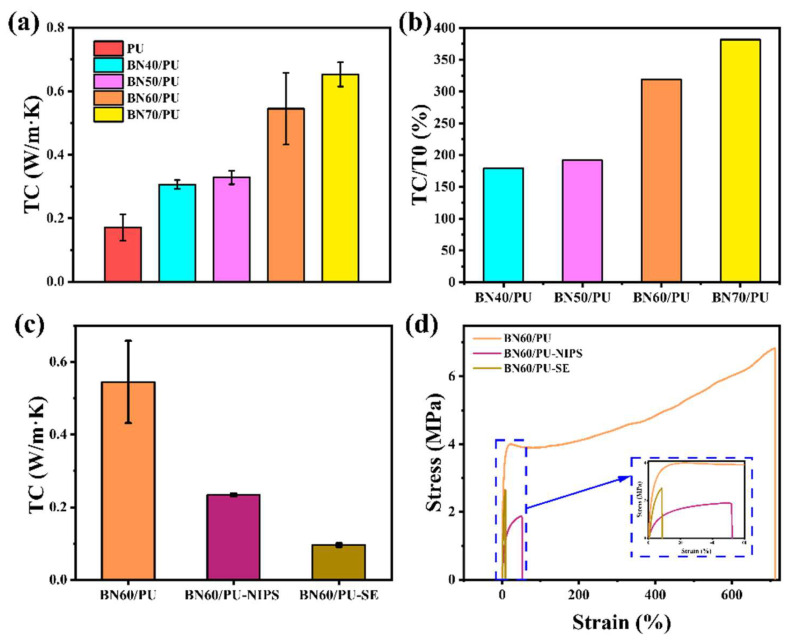
(**a**) TC of PU and BN/PU composite films; (**b**) TC enhancement of BN/PU composite films; (**c**) TC of BN60/PU, BN60/PU-NIPS, and BN60/PU-SE composite films; (**d**) Stress–strain of BN60/PU, BN60/PU-SE, and BN60/PU-NIPS composite films, respectively.

**Figure 5 ijms-24-08221-f005:**
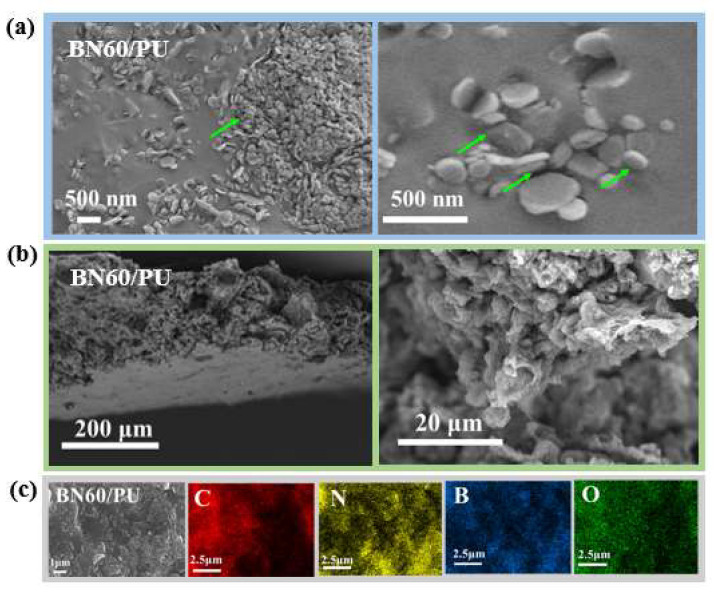
(**a**) Freeze-fractured and (**b**) tensile-fractured morphologies of BN60/PU composite film; (**c**) EDS elemental mapping of freeze-fractured BN60/PU composite film.

**Figure 6 ijms-24-08221-f006:**
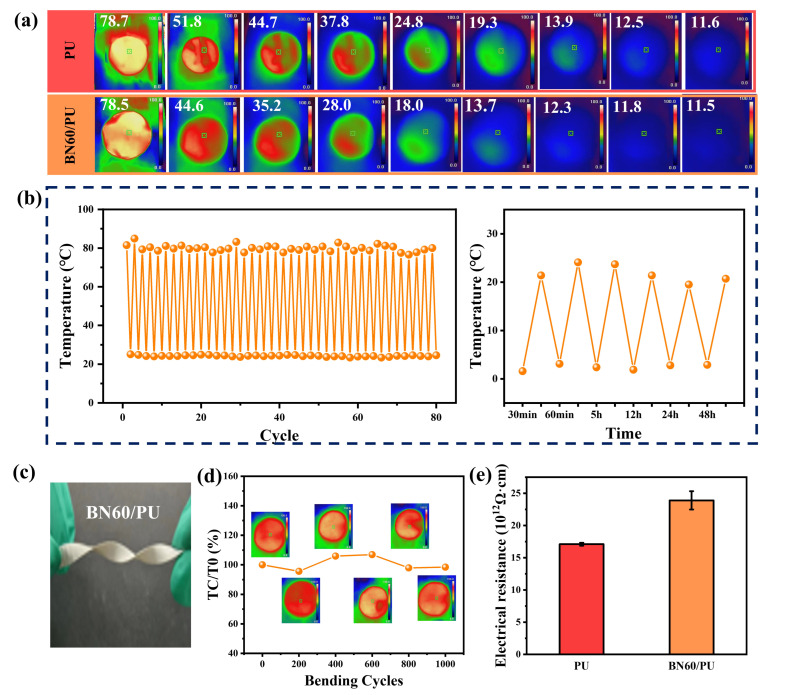
(**a**) Infrared thermal images and surface temperature of PU and BN60/PU composite film at different times; (**b**) Cyclic heating and cooling tests showing good stability of BN60/PU composite film; (**c**) Photograph of flexible BN60/PU composite film; (**d**) Surface temperature of BN60/PU composite as functions of bending cycles; (**e**) Insulating property of PU and BN60/PU composite film.

**Table 1 ijms-24-08221-t001:** TGA data of PU and BN/PU composite films under nitrogen condition.

Sample	T_5%_ (°C)	T_20%_ (°C)	Char Residue (%)
PU	281.5	313.5	2.30
BN40/PU	297.6	355.4	35.38
BN50/PU	284.8	334.9	49.12
BN60/PU	302.5	360.4	60.00
BN70/PU	306.0	398.3	64.92
PU	281.5	313.5	2.30

## Data Availability

Not applicable.

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
