# Peer review of "Boron Nitride/Polyurethane Composites with Good Thermal Conductivity and Flexibility"

_ijms, 2023, doi:10.3390/ijms24098221_

Round 1

Reviewer 1 Report

In this manuscript, the authors fabricated h-BN/PU composites using a novel synthesis routine and demonstrated their thermal and mechanical properties. They have conducted a large variety of experiments to provide adequate evidence of the potential applications in future electronics with their materials, and they exhibited the results in an understandable and scientific manner. Therefore, we recommend that the manuscript be accepted after addressing the following revision requests:

1. There are previously published works of similar material class with better thermal properties, such as [DOI: 10.1016/j.compscitech.2018.03.028]. The authors shall do more thorough literature review and discuss how their samples are better than existing works.

2. The authors studied stress and strain dependence on the h-BN content. Before the measurement, did the authors released strains left from the synthesis process? This is usually done by sample transfer or sample annealing, and we assume this process to be important for isolating inherent strains from material growth effects.

3. The mentioned Fig. 5f in line 246 does not exist.

The English used in the manuscript is generally readable and professional, yet the authors are still suggested to carefully revise certain wordings and terms. Taking page 1 as an example, note the expressions "Thermally insulating composites" in line 14, "insulating properties" in line 16, "amorphous BN" instead of "unordered BN" in line 21, and "thermally conductive composites" in line 30.

Author Response

  1. There are previously published works of similar material class with better thermal properties, such as [DOI: 10.1016/j.compscitech.2018.03.028]. The authors shall do more thorough literature review and discuss how their samples are better than existing works.

Response:

We would like to thank the reviewer for the careful and thorough reading of this manuscript and for the thoughtful comments and constructive suggestions, which helped us to improve the quality of this manuscript. In this study, they report a novel approach to fabricate large area, high-filler-loaded, flexible and insulating hBN/thermoplastic polyurethane (TPU) composite films via solution ball-milling of hBN and TPU. Flexible BN/PU composite films were successfully fabricated in their research[1]. Niu et al. presented here is a close-stack thermally conductive three-dimensional (3D) hybrid network structure prepared by a simple and green strategy(solution casting). The similar point between these articles was to form a horizontally arranged network within the polymer matrix or the preparation step[2]. Nanocomposites based on graphene oxide (GO), hexagonal boron nitride (h-BN), and their hybrid GO/h-BN as nanofillers in polyurethane (PU) were prepared by solution casting. An impressive improvement of up to ~1450% in thermal conductivity was observed for the same sample when compared to neat polymer[3]. The above researchs expressed that the polymer filled with fillers by solution could obtain uniform and stable composites. Thanks again for the your constructive guidance on the article. Base on the tranditioal solution casting method, we improved the NIPS process to get harmonious composite solution, and then the brinary solvent was benfiical to the casting process. Finally, the BN60/PU composite formed by the improved NIPS method express excellent mechanicla preperty(7.52±0.87 MPa stress and 707.34±38.34%strain). The more details have been added in the manuscript.(Section 1)

“Yu et al.[13] fabricate flexible and insulating elastomer-based 2D composite film with excellent TC through ball milling and hot-pressing. The hot-compression induced alignment of h-BN and strong interactions in the polymer matrix, improving the thermal conductivity of the composites.”

  1. The authors studied stress and strain dependence on the h-BN content. Before the measurement, did the authors released strains left from the synthesis process? This is usually done by sample transfer or sample annealing, and we assume this process to be important for isolating inherent strains from material growth effects.

Response:

We are very grateful to you for the constructive advices. First of all, all the samples in our work were not subjected to strain release. What’s more, NIPS method was usually used in polymer casting. By NIPS, we made LIG directly on uncoated porous PES membranes, while preserving the subsurface polymer. A self-healing coating containing polyethyleneimine (PEI)/ammonium polyphosphate (APP) polyelectrolyte complex was successfully prepared by NIPS, and then was deposited on flexible polyurethane foam (FPUF) in association with graphene oxide (GO) via a simple dip-nip process to improve the flame retardancy and mechanical property with no more strain release[4]. An endeavored to design three types of CNTs based thermoplastic polyurethane (TPU) composites through a facile solution blending with non-solvent induced phase separation (NIPS) strategy with nomore strain release[5]. The above articles proved that the polymer casted by NIPS method could get excellent property and wide application with no more strain release.what’s more, in this paper, PU chips were directly mixed with BN, and the inproved NIPS process is adopted to promote the formation of the mixed films. Then, according to the characteristics of the sample, the performance of the composite film prepared by traditional solvent evaporation method and non-solvent phase transition method is compared. The hot pressing process is mainly to study the effect of the arrangement of BN particles on the performance of the composite film. And the hot pressing temperature did not reach the glass transition temperature of PU resin, which did not have the effect on annealing treatment on PU. Thanks again for your constructive guidance on the manuscript!

  1. The mentioned Fig. 5f in line 246 does not exist.

Response:

Thanks very much for your advice! I'm sorry we made an oversight in the manuscript review. And we have adjusted the description the Section 2.6 in the manscript. Othewose, we have adjusted the graph in the mnuscript.in section 2.

“As shown in Fig. 6a, the surface temperature of each sample decreased with time…

Additionally, Fig. 6b indicates that the composite film still has good thermal conduc-tivity after 48 h of standing at low temperature.…

Fig. 6c shows flexible of composite film. Moreover, the BN60/PU maintained good integrity and showed a negligible decrease in the surface temperature after 1000 cycles of bending, thus demonstrating good mechanical and thermal reliability in practical use (Fig. 6d)….

As shown in Fig. 6e,”

Reviewer 2 Report

This manuscript is research on “Boron nitride/polyurethane composites with good thermal conductivity and flexibility”.  I would like to say that the research is quite impressive, unourtunately unacceptable without major revision. comments:

1.    Please explain in detail all the chemicals used, referring to the properties of the auxiliary chemicals.

2.    Describe in more detail the experimental steps in the synthesis of PU/NIPS and BN60/PU-NIPS composites and please cite

3.    As seen in the XRD spectra, an intercalated structure is observed between the BN/PU composites. Please let the authors comment on the advantages and disadvantages of intercalated or exfoliated structures.

4.    Why d001 peaks were not observed in XRD analysis results need explanation

5.    When the analysis results were compared, BN60/PU composites gave more impressive results in general. However, this difference between 70% RH added composites needs to be interpreted in more detail.

6.    Please explain the spectra in the FTIR analysis results in more detail and cite the article. The research below will help you. The quote below will give you an idea.

“The FTIR spectra for the PU composites are shown in Figs. 5, 6 and 7. The combination of NH deformation and CN stretching vibration (amide II bands) occurred at 1521 cm-1 . The absorption band resulting from carbonyl stretching vibration (amide I band) of the urethane prepolymer appeared at around 1726 cm-1 . The absorption band at 2273 cm-1 , attributed to free isocyanate groups (N] C] O), confirmed that there was an excess of NCO-terminated groups in the polymer [19]. The FTIR spectra of PU are shown in Figs. 5, 6 and 7; these spectra indicate formation of urethane groups by presence of absorption peaks at 3340 cm-1 [20]. Furthermore, as seen in Figs. 5, 6 and 7, the characteristic absorptions of the PU were observed at 2276 cm-1 (N=C=O asymmetric stretching), 1711 cm-1 (C=O stretching of N-aryl urethane), 1509 cm-1 (in-plane N–H bending), and 1092 cm-1 (C–O stretching) [21, 22].”

Improvement of Synthesis and Dielectric Properties of Polyurethane/Mt-QASs+ (Novel Synthesis)

G Baysal, H Aydın, H Hoşgören, S Uzan, H Karaer

Journal of Polymers and the Environment 24, 139-147

7.    Minor editing of English language required

Minor editing of English language required

Author Response

1.Please explain in detail all the chemicals used, referring to the properties of the auxiliary chemicals.

Response:

Thanks very much for your advice! I'm sorry we left out some of the chemical details. And the whole chemicals details were addede into the the manscript(Section 3.1).

“Hexagonal boron nitride (BN, 1-2 μm) was purchased from Macklin Biochemical Co., Ltd. The thermoplastic polyurethane (PU, 1185A) was provided by BASF Co., Ltd. N, N-dimethylformamide (DMF, 99.5 %, AR) and toluene (TL, 99.5 %, AR) were purchased from Sinopharm Chemical Reagent Co., Ltd. All chemicals were used di-rectly without further treatment. The coagulating bath was made by deionized water (DI water, conductivity ≤ 16 MΩ·cm).”

2.Describe in more detail the experimental steps in the synthesis of PU/NIPS and BN60/PU-NIPS composites and please cite

Response:

Thanks very much for your suggestions! Base on the tranditional NIPS process, we improved it to obtain  excellent thermal conductivity PU/BN composite. For tranditional NIPS, Yang et al. prepared the polyvinylidene difluoride (PVDF) lithium battery (LIB) separators by NIPS technique, and the composite films showed the formation of long finger-like pore structure[5]. Futhermore, the the vapor-induced phase separation coupled with non-solvent induced phase separation (VIPS-NIPS method) could achieve the purpose of preventing the formation of macro-voids and enhancing the mechanical stability. In summary, we improved the NIPS method to obtain the mechanical stability conposite films based on the above research. Thanks again for the reviewer's suggestion. Based on the literature research, the experimental steps were shown in detail in the manscript(Section 3.2).

“Fig. 1a illustrates the preparation of BN/PU composite films using improved non-solvent induced phase separation (NIPS) process using binary solvents. Firstly, the calculated BN and PU was dissolved in binary solvents (DMF:TL=1:1 wt%) to prepare a 30 wt% casting solution, the solution was cast onto a clean glass plate model, and the model was quickly moved into DI water coagulation bath for 120 min. After that, the formed film was dried at 80 ℃ for 1 h. Finally, the as-prepared film was pressed under 5 MPa pressure at 100 ℃ for 3 min. The PU with 0, 40, 50, 60, and 70 wt% BN content were coded as PU, BN40/PU, BN50/PU, BN60/PU, and BN70/PU, respectively.

The PU filled with 60 wt% BN formed by traditional NIPS method was named as BN60/PU-NIPS, according to previous publication.[31] The PU was firstly dissolved in DMF, and the mixture was cast on a glass plate, immersed in DI water for 120 min. After that, the formed film were dried at 80 ℃ for 1 h. Finally, the composite film was pressed under 5 MPa pressure at 100 ℃ for 3 min.

The PU filled with 60 wt% BN formed by traditional solvent evaporation (SE) method was named as BN60/PU-SE.[32] The composite film was prepared by the following steps. PU was dissolved in DMF, and the mixture was cast on a glass plate, dried at 80 ℃ for 1 h. After that, the formed film was pressed under 5 MPa pressure at 100 ℃ for 3 min.”

3.As seen in the XRD spectra, an intercalated structure is observed between the BN/PU composites. Please let the authors comment on the advantages and disadvantages of intercalated or exfoliated structures.

Response:

We are very grateful to you for the constructive guidance! The BN/PU composites with intercalated structure were formed by h-BN and PU, and the exfoliated structures were constructed by BNNs which were peeled by h-BN. What’s more, h-BN has been an appropriate candidate for thermal conductivity applications due to its outstanding thermal conductivity (600 W∙m−1∙K−1), excellent oxidation resistance, anisotropic properties, and high temperature resistance. Although researchers have applied h-BN individually to enhance the thermal performance of PU[6], problems with aggregation and poor thermal cycling stability remain. As a typical 2D material, boron nitride nanosheets (BNNSs) display the intrinsic TC with high value and strong anisotropy (the in-plane and through-plane TC is about 600 and 2–30 W/mK, respectively). The synthetic routes of BNNSs include chemical vapor deposition [7], micromechanical cleavage of BN platelets [8], and unzipping of BN nanotubes [9]. Although these approaches yield relatively high quality BNNSs, the low production and high cost significantly restrict their scalability. So we used the h-BN to construct intercalated structure in the BN/PU composite films by the improved NIPS method, which could effectively enhanced the interface force between polymer and inorganic fillers in our preious researchs. The points haved added into the manuscript(Section 1).

“Compared with BN, boron nitride nanosheet (BNNS) has higher thermal conductivity, but it needs to be peeled off through various methods, such as ball milling, ultrason-ic-assisted exfoliation with different surface modifier. Although these approaches yield relatively high quality BNNSs, the low production and high cost significantly restrict their scalability. Based on the principle of simplicity and mass production, the interca-lated BN is selected for this study.”

4.Why d001 peaks were not observed in XRD analysis results need explanation

Response:

We are very grateful to you for the constructive advices, and we apologize for the vague expressions on the XRD of PU and PU/BN composites. And we have modified in Fig.1b, whole characteristic crystalline peaks of BN were added into it. What’s more, the description about the XRD pattern of the BN was modified in manuscript(Section 2.1).

“As shown in Fig.1b-c, the XRD pattern of the BN reveals the characteristic diffraction peaks at 26.7, 41.7 43.9, 50.2 and 55.1°, which correspond to the (002), (100), (101), (102) and (004) planes……”

5.When the analysis results were compared, BN60/PU composites gave more impressive results in general. However, this difference between 70% RH added composites needs to be interpreted in more detail.

Response:

We are very grateful to you for the constructive advices. For BN/PU composite films, the the stress and strain of BN60/PU are 7.52±0.87 MPa and 707.34±38.34%, respectively. When the content of BN particles was 60%, the sample expressed the excellent mechical property. However, When the BN content continued to increase to 70%, the stress and strain of the BN70/PU are 5.00±0.50 MPa and 274.67±44.74%, respectively, and the strain of the composite film decreased significantly. That’s reason why we chose BN60/PU composites to do more research in detail. Beyond that, for the termal property, the char residue of BN60/PU and BN70/PU was 60.00, 64.92%,respectively. For the BN60/PU, the char residue was close to the fillers content, but BN70/PU did not. It could be conclude to agglomerate when its content is 70%. And we thanks for your constructive suggestions to my research, more details of BN70/PU have been added in the manuscript.

Section 2.3

“When the BN content continued to increase to 70 wt%, the stress and strain of the BN70/PU are 5.00±0.50 MPa and 274.67±44.74%, respectively, and the strain of the composite film decreased significantly.”

Section 2.4

“However, the residual mass of BN70/PU is only 64.92%, implying that the high content of BN is severely agglomerated, which disturbs the network of PU and makes the stress concentration.”

Line 166-168

6.Please explain the spectra in the FTIR analysis results in more detail and cite the article. The research below will help you. The quote below will give you an idea.

“The FTIR spectra for the PU composites are shown in Figs. 5, 6 and 7. The combination of NH deformation and CN stretching vibration (amide II bands) occurred at 1521 cm-1 . The absorption band resulting from carbonyl stretching vibration (amide I band) of the urethane prepolymer appeared at around 1726 cm-1 . The absorption band at 2273 cm-1 , attributed to free isocyanate groups (N] C] O), confirmed that there was an excess of NCO-terminated groups in the polymer [19]. The FTIR spectra of PU are shown in Figs. 5, 6 and 7; these spectra indicate formation of urethane groups by presence of absorption peaks at 3340 cm-1 [20]. Furthermore, as seen in Figs. 5, 6 and 7, the characteristic absorptions of the PU were observed at 2276 cm-1 (N=C=O asymmetric stretching), 1711 cm-1 (C=O stretching of N-aryl urethane), 1509 cm-1 (in-plane N–H bending), and 1092 cm-1 (C–O stretching) [21, 22].”

Improvement of Synthesis and Dielectric Properties of Polyurethane/Mt-QASs+ (Novel Synthesis)

G Baysal, H Aydın, H Hoşgören, S Uzan, H Karaer

Journal of Polymers and the Environment 24, 139-147

Response:

We would like to sincerely thank you for your constructive advices! The FTIR analysis in this articles was clear logic, simple sentences and detailed description, which is worth my careful study. Base on this, the description of FTIR analysis results was rewriten in the manuscript (Section 2.2).

“The characteristic absorptions of the PU were observed at 3312 cm-1 (N-H stretching vibration), 2939 cm-1 (asymmetric stretching vibration of -CH2-), 2853 cm-1 (symmetric stretching vibration of -CH2-), 1730 cm-1 and 1700 cm-1 (stretching vibration of C=O), 1531 cm-1 (N-H bending vibration), 1413 cm-1 (bending vibration of -CH2 and -CH3), 1222 cm-1 (C-N stretching vibration), and 1075 cm-1 (stretching vibration of C-O-C).[30]”

7.Minor editing of English language required

Thanks very much for your advice! Some grammatical mistakes have been amended.

Line 69“Herein, binary solvents were employed to improve the interfacial interaction be-tween the BN and PU matrix,”

Round 2

Reviewer 2 Report

Dear Authors, 

I am pleased to inform that the revised manuscript has been accepted.